# A comprehensive benchmark of graph neural networks, graph kernels, and classical machine learning approaches on rs-fMRI brain graphs

**Razan Mhanna**[1,2] iD        RAZAN.MHANNA@INRIA.FR

**Sophie Achard**[1] iD        SOPHIE.ACHARD@INRIA.FR

**Alexander Petersen**[3]        PETERSEN@STAT.BYU.EDU

**Jonas Richiardi**[4] iD        JONAS.RICHIARDI@CHUV.CH

[1] *Univ. Grenoble Alpes, CNRS, Inria, Grenoble INP, LJK, Grenoble, France*

[2] *Univ. Grenoble Alpes, Inserm U1216, CHU Grenoble Alpes, Institut des Neurosciences, France*

[3] *Department of Statistics, Brigham Young University, Provo, UT, USA*

[4] *Lausanne University Hospital and University of Lausanne, Switzerland*

**Editors:** Accepted for publication at MIDL 2026

## Abstract

Resting-state functional MRI (rs-fMRI) provides a powerful lens through which large-scale brain organization can be examined by modeling functional connectivity as a graph. These functional brain graphs now form the basis of machine-learning applications in neuroscience, ranging from relatively straightforward classification problems to more challenging behavioral and cognitive prediction tasks. While graph neural networks (GNNs) have gained increasing attention in neuroimaging, the absence of a unified, reproducible benchmark comparing GNNs with classical machine-learning models and graph kernel methods, across heterogeneous datasets and tasks, has made it difficult to assess their relative strengths. In this work, we introduce a comprehensive benchmarking framework spanning four heterogeneous cohorts ($N = 1513$) and multiple classification tasks, including clinical diagnosis and phenotypic prediction. We systematically evaluate classical models, graph kernels, and representative GNN architectures under a rigorous repeated nested cross-validation design and assess pairwise differences using the corrected repeated k-fold test with false-discovery-rate control. Our results show that, for this class of relatively small graphs with fixed vertex ordering, well-tuned classical ML approaches and graph kernels are competitive with GNNs, while requiring substantially fewer computational resources. For instance, the Shortest-Path graph kernel achieves 0.98 accuracy on the COMA dataset, logistic regression reaches 0.81 accuracy and 0.63 MCC on HCP sex prediction, and all model families cluster closely on multi-site datasets such as ABIDE and ADHD, where no statistically significant differences emerge. All code, seeds, cross-validation folds, fold-specific hyperparameters, full prediction logs and computational-cost measurements are publicly released at https://gitlab.inria.fr/rmhanna/benchmark-study to ensure full transparency and reproducibility. This benchmark provides practical guidance for model selection in rs-fMRI connectome analysis.

**Keywords:** Resting-state fMRI, brain networks, Graph kernels, Graph neural networks, benchmarking, reproducibility, computational efficiency.

## 1. Introduction

A popular approach used to investigate brain function is resting-state functional magnetic resonance imaging (rs-fMRI), a noninvasive neuroimaging technique that measures fluctuations in the blood-oxygenation-level-dependent (BOLD) signal as a correlate of brain activity. fMRI data inherently possess a complex spatio-temporal structure: each voxel (volumetric pixel) provides a time series of BOLD measurements, resulting in high-dimensional, noisy observations that inherently reside on non-Euclidean domains. To reduce the spatial complexity, voxels can be collated into user-specified regions of interest (ROI), after which aggregation is performed across voxels within the same ROI by averaging. Based on these regional signals, a functional connectome is commonly modeled as a graph, where nodes correspond to ROIs and edges represent estimated pairwise functional connectivity. Such graph-based representations have been successfully applied across a broad spectrum of network neuroscience studies, including investigations of neurodegenerative conditions such as Alzheimer's disease and mild cognitive impairment, autism spectrum disorder, disorders of consciousness, and the prediction of psychometric, cognitive, and behavioral phenotypes in healthy individuals (Dadi et al., 2019; Di Martino et al., 2014; Li et al., 2021; Cui et al., 2022). However, despite substantial progress in neuroimaging analysis and machine learning, significant challenges persist due to the intrinsic complexity of functional connectivity data—including high dimensionality, sensitivity to parcellation schemes, variability across acquisition sites, and limited sample sizes. As a result, developing reliable computational methods that are robust, generalizable across cohorts, and reproducible remains an open and pressing problem in functional connectivity modeling.

Prior benchmarking efforts in brain functional connectivity have mainly focused on either GNNs or classical ML pipelines. BrainGB (Cui et al., 2022) evaluates message-passing GNN architectures across three functional datasets (HIV disease classification, PNC and ABCD gender prediction) and one structural dataset (PPMI Parkinson's disease classification), using a modular design to compare a large family of GNN operators. In parallel, the benchmark of Dadi et al. (2019) systematically assessed classical ML models over 6 datasets, 8 atlases, and 3 connectivity profiles, highlighting the strong performance of tangent-space parametrization and $\ell_2$-penalized classifiers. We focus on these two benchmarks because they represent the most comprehensive and methodologically rigorous evaluations currently available: BrainGB provides the most extensive assessment of GNN architectures, whereas Dadi et al. (2019) remains the reference point for classical ML pipelines. Other studies typically investigate a single dataset, a single prediction task, or a narrowly defined model family, making broad methodological comparisons difficult. Nonetheless, these benchmarks remain limited to either GNNs or shallow ML, and do not provide a unified comparison spanning ML, graph kernels, and GNNs across heterogeneous datasets and tasks.

In this work, we develop a systematic benchmark that integrates multiple cohorts with distinct demographic and clinical profiles, including ADHD, HCP, ABIDE, and a clinical COMA dataset. This multi-cohort design allows us to assess model performance across a broad range of prediction tasks: clinical case–control classification, phenotypic sex prediction, ASD diagnosis, and ADHD classification, mirroring the diversity of applications commonly encountered in network neuroscience. To the best of our knowledge, no prior study has provided a unified comparison of classical machine-learning models, graph-kernel

methods, and graph neural networks across such heterogeneous datasets and task paradigms. Throughout this study, we focus exclusively on inductive GNNs for graph–level prediction, where each subject is represented by an independent brain graph; transductive population-level GNNs for node classification are naturally excluded.

Our contributions are threefold:

- Multi-dataset, multi-task evaluation: We evaluate models acrosss four cohorts covering both healthy and clinical populations, enabling assessment on diverse classification tasks (clinical diagnosis, ASD/ADHD classification, sex prediction).

- Comprehensive comparison of modeling families: We benchmark classical ML methods (logistic regression, SVM, XGBoost), graph–kernel approaches (Shortest–Path, Weisfeiler–Lehman), and GNN architectures (GCN, GraphSAGE, GAT) under a unified and fully reproducible experimental pipeline.

- Robust evaluation strategy: We employ a repeated nested cross-validation framework in which, for each of the 50 outer folds, one fold is reserved exclusively for testing, whereas the remaining folds are further partitioned into training and validation subsets. This procedure has been demonstrated to provide more stable model rankings (Eve et al., 2025).

- Statistical significance analysis: We assess whether performance differences between models are statistically significant using the corrected repeated k-fold test of Bouckaert and Frank (2004), combined with Benjamini and Hochberg (1995) FDR correction test across pairwise comparisons.

- Practical efficiency assessment: We report computational aspects including training time, and $CO_2$ emisions, for CPU-operated classical machine learning and graph kernels models. This provides a clear picture of the computational footprint of lightweight methods within our benchmark.

Taken together, this benchmark offers a transparent and systematic comparison spanning classical machine-learning models, graph-kernel methods (which remain underexplored in the neuroimaging community) and more recent GNN architectures, evaluated across multiple datasets and prediction tasks. Our goal is to provide a clearer picture of the relative strengths of each modeling family in terms of accuracy, robustness, and computational efficiency.

## 2. Materials and Methods

### 2.1. Datasets

In this study, we selected publicly available resting-state fMRI datasets that offer comparable preprocessing pipelines and compatible brain parcellations, enabling methodological coherence across datasets. This choice allows us to assess the generalizability of state-of-the-art (SOTA) models under varying sample sizes, scanner characteristics, and population demographics. We first use data from the publicly available, fully de-identified Neuro

| Dataset | Source | Atlas | #ROIs | Classification task | # Subjects |
|---------|--------|-------|-------|--------------------|-----------|
| HCP | Dadi et al. (2019) | AAL | 116 | Sex classification | 443 |
| ADHD | Bellec et al. (2017) | AAL | 116 | ADHD prediction | 160 |
| ABIDE | Dadi et al. (2019) | AAL | 116 | ASD prediction | 866 |
| COMA | Oujamaa et al. (2023) | AAL | 105 | DoC prediction | 44 |

Table 1: Summary of dataset statistics. ADHD = Attention Deficit Hyperactivity Disorder; ASD = Autism Spectrum Disorder; DoC = Disorders of Consciousness.

Bureau ADHD-200 dataset (Bellec et al., 2017), which originally includes participants recruited across eight sites: Peking University, Bradley Hospital (Brown University), Kennedy Krieger Institute, the Donders Institute, New York University Child Study Center, Oregon Health and Science University, the University of Pittsburgh, and Washington University in St. Louis. All cohorts received approval from their respective institutional review boards, and written informed consent was obtained from all participants or their legal guardians. Individuals had no history of psychiatric, neurological, or medical conditions other than ADHD.

We also include the ABIDE dataset, released through the Autism Brain Imaging Data Exchange initiative (Di Martino et al., 2014), which aggregates rs-fMRI acquisitions from multiple sites to study Autism Spectrum Disorder. We rely on rs-fMRI time series provided by the Preprocessed Connectome Project (PCP) (Craddock et al., 2013), as used in Dadi et al. (2019), and use them to perform ASD vs. control classification. Then, we incorporate data from the Human Connectome Project (HCP), which offers high-quality imaging and behavioral assessments for healthy young adults (Van Essen et al., 2013). We use the preprocessed rs-fMRI time series from the HCP900 release, also recovered from the preprocessing distributed in Dadi et al. (2019). This dataset enables experiments on sex classification, providing a complementary setting with substantially longer acquisitions. Finally, we use a clinical dataset of 44 subjects acquired at Grenoble Alpes University Hospital (Oujamaa et al., 2023). This cohort comprises 24 patients who had sustained acute severe traumatic brain injury; at the time of scanning, 15 had recovered consciousness, whereas 9 remained in a minimally conscious state (MCS). A control group of 20 age-matched healthy volunteers was collected under comparable acquisition conditions (Job et al., 2020). All rs-fMRI scans were obtained on the same MRI system using identical acquisition parameters and were parcellated using a 105-region modified AAL3 atlas.

A summary of all datasets, associated prediction tasks, and sample sizes is provided in Table 1.

## 2.2. Graph Construction

For each subject, regional BOLD time courses were collected from preprocessed datasets reported in Section 2.1. Based on these signals, pairwise Pearson correlation coefficients were computed using the ConnectivityMeasure function from the Nilearn library (Abraham et al., 2014), resulting in one weighted connectivity matrix per subject. Depending on the model requirements, different graph representations were derived from these matrices.

For graph-kernel methods described in 2.3.2, we constructed thresholded binary graphs by first extracting the minimum spanning tree (MST) to ensure graph connectivity, and subsequently retaining the top 10% of the strongest remaining correlations. This threshold was selected based on empirical performance evaluation on the COMA dataset. In contrast, classical machine-learning models described in 2.3.1 were applied directly to the vectorized upper triangular part of the full, weighted connectivity matrices. For GNN-based models, we followed the BrainGB (Cui et al., 2022) recommendations by using connection profiles as node features, which have been shown to be among the most effective node feature choices for rs-fMRI brain connectome analysis. Under this formulation, explicit edge thresholding plays a less critical role, as connectivity information is primarily encoded in the node features rather than through a sparsified graph topology.

### 2.3. Benchmark Models

#### 2.3.1. Classical ML

As baseline classifiers, we employ three widely used machine-learning algorithms: Logistic Regression, Random Forests, and XGBoost. Contrary to many prior neuroimaging studies, including the benchmark of Dadi et al. (2019), we perform systematic hyperparameter optimization independently for each dataset. The full search space and optimization details are reported in Section 2.5. These classical models have consistently demonstrated strong performance across multiple domains, including neuroimaging studies.

#### 2.3.2. Graph kernels

Kernel methods provide a powerful mathematical framework for measuring similarities between structured objects, such as graphs, by implicitly mapping them from the original feature space to a (possibly infinite-dimensional) Hilbert space, where it corresponds to an inner product between transformed samples. Their main advantage lies in their computational efficiency: many kernels admit closed-form expressions for these inner products, removing the need to compute the explicit transformation. As established by Mercer's theorem, a function qualifies as a valid kernel if it satisfies the conditions of positive semi-definiteness, ensuring it represents a legitimate inner product in some feature space. Formally, for a non-empty set $\chi$ and a function $k : \chi \times \chi \to \mathbb{R}$, there exists a Hilbert space $\mathcal{H}_k$ and a feature map $\phi : \chi \to \mathcal{H}_k$ such that

$$k(x, y) = \langle \phi(x), \phi(y) \rangle_{\mathcal{H}_k}, \quad x, y \in \chi.$$

Once the kernel function is defined, kernel-based algorithms such as the Support Vector Machine (SVM) can be applied directly for classification or regression tasks (Hofmann et al., 2008).

In this study, we evaluate two graph kernels implemented in the GraKeL library (Siglidis et al., 2020)—the Shortest-Path and Weisfeiler–Lehman kernels.

- The Shortest-Path kernel measures graph similarity by comparing the lengths and endpoint labels of all shortest paths between node pairs. For each graph, the shortest-path distances are computed, and two graphs are considered similar when their node pairs are connected through paths of comparable lengths. This kernel captures the

overall topological layout of a network and is particularly effective when global path structure differentiates the graphs. It is later referred to as $GK - SP$ throughout the paper.

- The Weisfeiler–Lehman kernel relies on the iterative node-label refinement procedure proposed in the WL test of graph isomorphism. At each iteration, a node's label is updated by combining its current label with those of its neighbors, generating progressively enriched node representations. By comparing graphs across multiple refinement steps, the kernel captures hierarchical structural information and neighborhood similarity. Its efficiency and scalability make it a strong baseline for graph classification tasks. It is referred to as $GK - WL$ in the rest of the paper.

### 2.3.3. Graph Neural Networks

Graph Neural Networks (GNNs) have gained significant attention in the field of network neuroscience (Cui et al., 2022; Li et al., 2021; Comparini et al., 2026; Xu et al., 2023, 2024) due to their ability to effectively model and analyze complex graph structures. In this work, all GNN models are considered within an inductive, graph-level learning framework, where each subject is represented by an independent brain graph and the prediction target is defined at the graph level (e.g., clinical diagnosis or phenotypic classification). Consequently, transductive GNN approaches commonly used for node classification on population graphs are not considered, as they are not well suited to the multi-subject graph classification setting addressed here.

Most modern architectures can be expressed under the message passing neural network (MPNN) framework, in which node representations are iteratively updated by aggregating information from their neighborhoods. A generic MPNN layer can be written as:

$$\mathbf{h}_i^{(l+1)} = \phi\left(\mathbf{h}_i^{(l)},\, \square_{j \in \mathcal{N}(i)} \psi\left(\mathbf{h}_i^{(l)}, \mathbf{h}_j^{(l)}\right)\right),$$

where $\mathbf{h}_i^{(l)}$ is the feature vector of node $i$ at layer $l$, $\psi$ and $\phi$ are learnable functions, and $\square$ denotes a permutation-invariant aggregation operator.

In this study, we focus on three representative GNN architectures widely used in many applications including brain graph analysis:

- Graph Convolutional Network (GCN): A baseline architecture that updates node representations by combining features from adjacent nodes through a predefined, normalized aggregation strategy (Jiang et al., 2019).

- Graph Attention Network (GAT): Incorporates an attention mechanism that adaptively weights neighboring nodes by learning how much each one should contribute during the feature aggregation process, allowing the model to emphasize the most informative interactions (Veličković et al., 2017).

- GraphSAGE (SAmple and aggreGatE): A sampling-based architecture that learns node embeddings by aggregating information from a fixed number of sampled neighbors, enabling inductive generalization to unseen nodes and graphs (Hamilton et al., 2017).

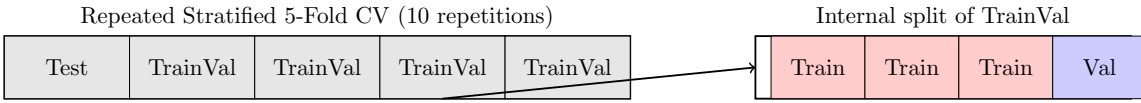

Figure 1: Repeated 5-fold nested cross-validation: Across 50 outer folds(5 folds repeated 10 times), one outer fold is held out as the test set, while the remaining four folds are internally split into three training folds and one validation fold for hyperparameter tuning.

- Graph Transformer (TransformerConv): An attention-based message-passing architecture that learns adaptive aggregation weights over neighboring nodes defined by the input graph topology, providing a flexible alternative to convolution-based GNNs Shi et al. (2021).

These architectures provide complementary mechanisms for learning from functional brain connectivity graphs and form suitable candidates for comparative evaluation in this study.

## 2.4. Evaluation metrics

The performance of our classification models is assessed using accuracy (ACC), balanced accuracy (BACC), and the Matthews correlation coefficient (MCC). While ACC quantifies the overall proportion of correctly classified samples, BACC accounts for class imbalance by averaging sensitivity and specificity. MCC provides a more comprehensive and balanced summary of performance by incorporating all four entries of the confusion matrix. It is defined as:

$$\text{MCC} = \frac{TP \times TN - FP \times FN}{\sqrt{(TP+FP)(TP+FN)(TN+FP)(TN+FN)}},$$

where $TP$, $TN$, $FP$, and $FN$ denote true positives, true negatives, false positives, and false negatives, respectively. MCC values range from $-1$ to $1$, with higher values indicating stronger agreement between predicted and true labels and offering a reliable assessment of classifier performance under class imbalance.

## 2.5. Implementation details and hyperparameter optimization

To minimize the risk of data leakage and obtain stable performance estimates on relatively small neuroimagaging datasets, all models were trained and evaluated within a repeated stratified K-fold cross-validation framework (10 repetitions, 5 folds). This choice is supported by recent empirical analyses demonstrating that repeated k-fold cross validation yields more reliable model rankings than single-split or non-repeated procedures, especially when sample sizes are limited within each outer fold (Eve et al., 2025). Hyperparameters were tuned independently using an internal stratified split of the training data, as illustrated in Figure 2.5, ensuring that the test fold remains completely unseen during both training and hyperparameter selection. The implementations were developed using PyTorch, NetworkX, PyTorch Geometric, GraKeL and CodeCarbon libraries (Fey and Lenssen, 2019;

Siglidis et al., 2020; Courty et al., 2024a). All experiments were conducted on a Dell work-station running Ubuntu 22.04, equipped with an Intel Core i7 processor and an NVIDIA RTX A500 GPU.

A systematic hyperparameter search was conducted to identify the optimal configuration for the classical machine-learning models and graph kernel classifiers. The search spaces were defined as follows: for Logistic Regression, the regularization parameter was $C \in [10^{-4}, 10^4]$ (log-uniform prior); for the Random Forest classifier, $n_{\text{estimators}} \in [100, 500]$, max_depth $\in [2, 20]$, min_samples_leaf $\in [1, 10]$, and max_features $\in \{"sqrt", "log2"\}$. For XGBoost, we optimized max_depth $\in [2, 6]$ and min_child_weight $\in [0.1, 10]$ (log-uniform prior).

Since the SP kernel produces a precomputed Gram matrix in GraKeL, only the SVM regularization parameter $C$ is tuned. For the Weisfeiler–Lehman (WL) kernel, we optimized both $C \in [10^{-3}, 10^3]$ and the WL height $h \in [2, 4]$. Hyperparameters were optimized separately for each dataset using Bayesian optimization on an inner validation split, with balanced accuracy as the criterion to maximize. Each search was limited to 20 Bayesian optimization calls. All seeds, fold-specific hyperparameters, and full logs are provided in our public GitHub repository to ensure transparency and reproducibility.

Due to the substantially higher computational cost of training GNNs compared to classical graph kernels, we adopt fixed hyperparameter settings that follow common practice in prior neuroimaging benchmarks, including BrainGB (Cui et al., 2022) and the work of Comparini et al. (2026). This approach is standard in multi-cohort evaluations, where exhaustive hyperparameter tuning for deep models is prohibitively expensive and provides limited benefit for small- to medium-sized neuroimaging datasets. All GNN models were trained with the Adam optimizer and a maximum of 200 epochs, with early stopping triggered after 30 consecutive epochs without improvement in the validation loss. For the GCN architecture, we used a learning rate of 0.001 with 32 hidden channels and three graph convolutional layers. The GAT model was configured with 16 hidden channels, four attention heads, and two layers, while GraphSAGE employed 32 hidden channels and three layers. All GNNs shared the same learning rate, number of epochs, optimizer, and early-stopping criterion

## 3. Results

### 3.1. Classification results

Table 2 summarizes the classification performance across all datasets in terms of accuracy (Acc) and Matthews correlation coefficient (MCC), while Figure 2 reports the corresponding balanced accuracy averaged over repeated cross-validation splits, with error bars indicating the standard deviation. Overall, we observe that classical machine learning methods, graph kernels, and graph neural networks achieve broadly comparable performance across cohorts. Importantly, graph kernels—particularly the Shortest-Path kernel (GK-SP)—achieve some of the highest performance levels, most notably on the COMA dataset, where GK-SP attains an accuracy of approximately 0.98 and an MCC of 0.97 (Table 2). On the ADHD dataset, GK-SP also yields competitive accuracy ($\approx$0.67), with lower MCC values, reflecting the difficulty of the task, likely due to the heterogeneity introduced by multi-site data acquisition. In addition, logistic regression achieves strong performance on both the HCP

Table 2: Classification performance across datasets. LR = Logistic Regression, RF = Random Forest, Acc = Accuracy, MCC = Matthews correlation coefficient.

| Model's family | Model | COMA | | HCP | | ADHD | | ABIDE | |
|---|---|---|---|---|---|---|---|---|---|
| | | Acc | MCC | Acc | MCC | Acc | MCC | Acc | MCC |
| Classical ML | LR | 0.84±0.1 | 0.71 ±0.19 | **0.81±0.03** | **0.63±0.07** | 0.62±0.07 | 0.11±0.17 | **0.63±0.03** | **0.26±0.06** |
| | RF | 0.8 ± 0.16 | 0.63±0.31 | 0.7±0.04 | 0.4±0.09 | 0.67±0.04 | 0.02±0.15 | 0.58±0.03 | 0.16±0.07 |
| | XGBoost | 0.71 ± 0.14 | 0.42 ± 0.35 | 0.73±0.03 | 0.46±0.07 | 0.62±0.06 | 0.01±0.17 | 0.6±0.03 | 0.19± 0.06 |
| Graph kernels | GK-SP | **0.98±0.04** | **0.97±0.07** | 0.71±0.04 | 0.41±0.09 | **0.67±0.03** | 0.04±0.16 | 0.6±0.04 | 0.19±0.08 |
| | GK-WL | 0.63±0.11 | 0.25±0.25 | 0.6±0.04 | 0.17±0.1 | 0.64±0.1 | 0.01±0.06 | 0.53±0.03 | 0.05±0.06 |
| GNN | GCN | 0.74 ± 0.15 | 0.49±0.32 | 0.68±0.05 | 0.35±0.1 | 0.61±0.1 | 0.1±0.2 | 0.53±0.04 | 0.06±0.07 |
| | GAT | 0.71±0.15 | 0.45±0.31 | 0.71±0.04 | 0.42±0.09 | 0.6±0.08 | 0.11±0.16 | 0.56 ± 0.03 | 0.13± 0.06 |
| | GraphSAGE | 0.75±0.13 | 0.52±0.27 | 0.73±0.04 | 0.45±0.09 | 0.64±0.07 | **0.16±0.17** | 0.57±0.04 | 0.13±0.08 |
| | TransformerConv | 0.79±0.12 | 0.6±0.24 | 0.73±0.05 | 0.45 ±0.09 | 0.62 ±0.07 | 0.14±0.17 | 0.57 ± 0.04 | 0.13 ± 0.07 |

(Acc ≈ 0.81, MCC ≈ 0.63) and ABIDE (Acc ≈ 0.63, MCC ≈ 0.26) datasets, highlighting the effectiveness of simple linear models in these settings. In contrast, graph neural network models, including GCN, GAT, GraphSAGE, and GraphTransformer, generally exhibit similar or slightly lower mean performance across datasets and cross-validation splits. This behaviour is also reflected in the balanced accuracy results shown in Figure 2, where GNNs typically match or underperform classical machine learning and graph kernel approaches.

The limited performance separation observed across the evaluated GNN architectures can be attributed to their shared reliance on message-passing and permutation-invariant aggregation mechanisms. Although the models differ architecturally, they all iteratively aggregate neighborhood information and propagate it across the graph. In the present setting, where graphs are relatively small and node identities are consistent across subjects, a small number of message-passing layers is sufficient to integrate near-global information. Under such conditions, increased architectural complexity or message-passing depth does not necessarily yield additional discriminative capacity and may instead lead to representation homogenization. Consequently, the clustered performance across GNN variants observed in this benchmark reflects not a deficiency of the models, but rather the characteristics of the data and task, under which the inductive biases introduced by message passing and aggregation have a limited differentiating effect.

The COMA dataset exhibits the clearest separation between model families, with graph kernels achieving the highest accuracy and MCC values, highlighting their ability to capture global alterations in functional brain connectivity associated with disorders of consciousness. In contrast, graph neural networks yield lower average performance in this setting, suggesting limitations in learning stable and discriminative representations from very small clinical datasets. As shown in Figure 2, balanced accuracy values are high overall but are accompanied by increased variability across cross-validation folds. This variability is likely related to the inter-patient heterogeneity within the COMA cohort, where patterns of functional alteration can differ substantially across individuals, leading to sensitivity to the specific composition of training and test folds. The HCP dataset exhibits the most homogeneous performance across models, with several approaches achieving similar accuracy values. Notably, logistic regression attains strong performance, comparable to or exceeding that of more complex graph-based models. This suggests that the high quality, consistency, and relatively low noise of the HCP data allow even simple linear models to capture

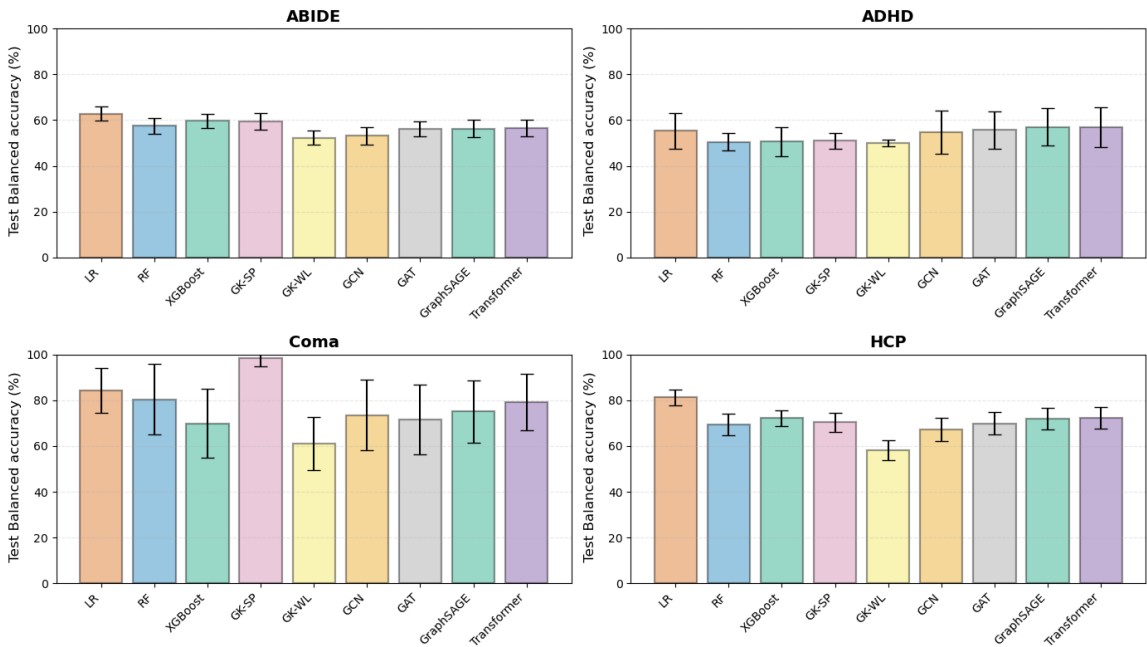

Figure 2: Repeated cross-validation balanced accuracy. The height of each bar corresponds to the mean balanced accuracy across all folds and repetitions, with error bars representing the standard deviation.

discriminative information effectively. For comparison with prior work, the NeuroGraph benchmark (Said et al., 2023) reports an accuracy of 69.9% for sex classification on the HCP Young Adult S1200 dataset (1,078 subjects) using a Random Forest classifier, which is broadly consistent with the performance of classical machine-learning baselines observed in our study. In contrast, graph neural networks in NeuroGraph achieve higher accuracy, with reported values of 75.46% for GCN, 77.69% for GraphSAGE, and 76.20% for GAT. However, these results are obtained under different experimental conditions, namely using static functional connectivity graphs constructed with the Schaefer atlas at 1000 ROIs and a single stratified 70/10/20 train—validation–test split, which limits direct comparability with our repeated cross–validation setting.

For the ADHD dataset, all methods exhibit relatively modest performance, with substantial overlap in balanced accuracy across model families, as shown in Figure 2. Graph kernels, particularly the Shortest-Path kernel (GK-SP), achieve some of the highest accuracy values ($\approx 0.67$ in Table 2). In terms of MCC, the best values are also attained by classical approaches, although MCC remains low overall across all models, reflecting the limited class separability in this dataset. This pattern reflects the subtle and heterogeneous nature of ADHD-related functional connectivity alterations, as well as the variability introduced by multi-site data collection. In this context, increased model complexity does not provide a clear advantage, and performance differences between approaches remain limited and often not statistically significant. A similar performance pattern is observed on the ABIDE

dataset, where the absence of clear gains from graph neural networks suggests that the high inter-site heterogeneity of ABIDE, combined with limited sample sizes per site, may hinder the learning of robust graph-level representations. In contrast, simpler models appear less sensitive to this variability, resulting in comparable or even superior performance. While careful hyperparameter tuning can modestly improve results, it does not fundamentally alter the observed patterns. In comparison with prior work, the ContrastPool study (Xu et al., 2024)reports, on the ABIDE dataset (989 subjects) evaluated using 10-fold cross-validation, an accuracy of $65.82 \pm 3.51\%$ for logistic regression and $61.18 \pm 5.01\%$ for random forest, values that are consistent with the performance of classical pipelines observed in our experiments. General-purpose GNNs achieve comparable or slightly lower accuracy, including GCN ($60.97 \pm 2.84\%$), GAT ($60.87 \pm 5.02\%$), and GraphSAGE ($63.09 \pm 3.11\%$). Importantly, GNNs in Xu et al. (2024) study benefit from explicit hyperparameter tuning via grid search.

### 3.2. Statistical comparison of models

To assess whether observed performance differences between models were statistically significant, we followed the recommendations of Bouckaert and Frank (2004), who showed that standard significance tests applied to cross-validation results can be overly optimistic due to dependencies between folds. We therefore employed the corrected repeated k-fold cross-validation test. For each dataset and each pair of models, predictions collected on identical test splits across all folds and repetitions were compared using a loss function derived from the evaluation metric. Statistical significance was then assessed using a two-sided Student's t-test on the mean loss difference. Furthermore, to account for multiple pairwise comparisons, Benjamini–Hochberg false discovery rate (FDR) correction was applied within each dataset (Benjamini and Hochberg, 1995). Raw p-values from the corrected repeated k-fold test are shown in the upper triangular part of each matrix, while FDR-adjusted p-values are reported in the lower triangular part of Figure 3. Starred cells indicate comparisons that remain statistically significant after FDR correction.

Across datasets, the statistical analysis largely supports the conclusion that most models achieve comparable performance, with limited evidence of robust pairwise differences. In particular, the ADHD dataset shows no statistically significant differences between models, as reflected by uniformly large p-values in both the raw and FDR-adjusted matrices. This lack of statistical separation is likely due to several factors, including the limited sample size retained from the ADHD-200 initiative (160 subjects in our study) and the multi-site heterogeneity, which results in small effective sample sizes per site. A similar pattern is observed for the ABIDE dataset, where only a small number of isolated pairwise comparisons remain significant after correction. As shown in Table 2 and Figure 2, most performance differences across models in ABIDE are small and statistically fragile, indicating substantial overlap between model families once cross-validation dependence and multiple testing are properly accounted for.

For the HCP dataset, the statistical comparisons reveal more consistent differences between model families than in ABIDE or ADHD. Logistic regression achieves the strongest performance, and several comparisons between LR and GNNs yield very small raw p-values in the upper triangular matrix (e.g., LR vs. GAT and LR vs. GCN with raw p-values $\approx$ 0.000), with some remaining significant after FDR correction. In addition, the Weisfeiler–

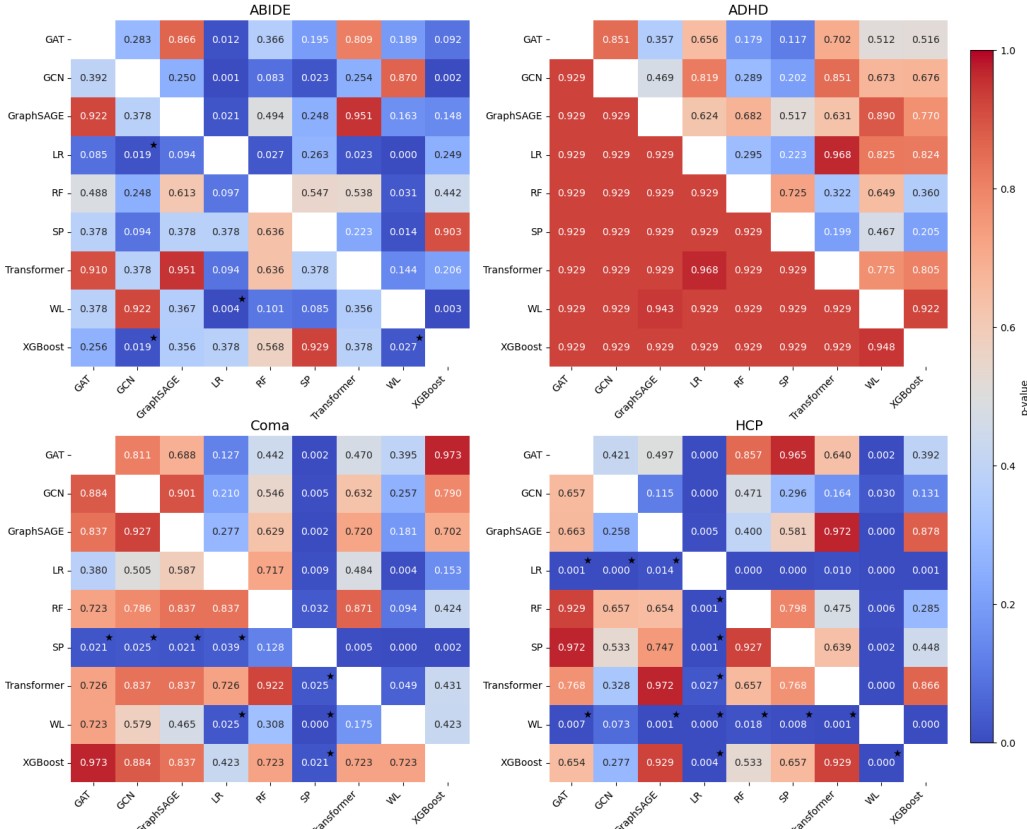

Figure 3: Pairwise statistical comparison of models across all datasets (ABIDE, ADHD, Coma, and HCP) based on accuracy. Each cell reports the p-value obtained from the corrected repeated k-fold cross-validation test (Bouckaert and Frank, 2004). Raw p-values are displayed in the upper triangular part of each matrix. The lower triangular part displays p-values after Benjamini–Hochberg false discovery rate (FDR) correction applied across all pairwise model comparisons within each dataset. Diagonal entries correspond to self-comparisons. Cells marked with a star indicate comparisons that remain statistically significant after FDR correction at level $\alpha = 0.05$.

Lehman (WL) graph kernel shows statistically significant differences when compared to most other models; however, these differences correspond to inferior performance, as WL consistently achieves the lowest accuracy and MCC values across models (Table 2). This indicates that, while WL captures graph representations that differ significantly from those learned by other approaches, these representations are less aligned with sex-related discriminative patterns in the HCP dataset. Overall, these results suggest that in a high-quality, homogeneous cohort such as HCP, simple linear models are sufficient to capture relevant population-level effects, whereas increased structural expressiveness—as in WL kernels or GNNs—does not necessarily translate into improved predictive performance.

Table 3: Total training time (seconds) and $CO_2$ emissions (kg), aggregated across all cross-validation folds and repetitions.

| Model | ABIDE | | ADHD | | COMA | | HCP | |
|---|---|---|---|---|---|---|---|---|
| | Time (s) | $CO_2$ (kg) | Time (s) | $CO_2$ (kg) | Time (s) | $CO_2$ (kg) | Time (s) | $CO_2$ (kg) |
| LR | 15 850.97 | $1.26{\times}10^{-2}$ | 1 499.40 | $5.09{\times}10^{-4}$ | 369.74 | $1.51{\times}10^{-4}$ | 8 202.85 | $6.76{\times}10^{-3}$ |
| RF | 7 916.05 | $1.27{\times}10^{-3}$ | 1 395.58 | $2.14{\times}10^{-4}$ | 386.46 | $5.95{\times}10^{-5}$ | 4 467.13 | $6.80{\times}10^{-4}$ |
| XGBoost | 450 497.07 | $2.94{\times}10^{-1}$ | 5 210.19 | $8.32{\times}10^{-4}$ | 559.06 | $1.14{\times}10^{-5}$ | 47 518.22 | $5.89{\times}10^{-3}$ |
| GK-SP | 4 537.77 | $6.97{\times}10^{-4}$ | 810.98 | $1.22{\times}10^{-4}$ | 228.32 | $3.47{\times}10^{-5}$ | 2 332.29 | $3.56{\times}10^{-4}$ |
| GK-WL | 9 429.97 | $1.38{\times}10^{-3}$ | 1 012.94 | $1.61{\times}10^{-4}$ | 230.97 | $3.49{\times}10^{-5}$ | 2 317.81 | $3.58{\times}10^{-4}$ |

In contrast, for the COMA dataset, the Shortest-Path graph kernel exhibits the strongest and most statistically consistent differences when compared with other model families, (e.g., raw p-values as low as $p < 0.02$ for SP versus GraphSAGE or GAT, with comparisons remaining significant after FDR correction at $p<0.05$), while differences among the remaining models are less pronounced. These results indicate that shortest-path–based features capture discriminative network patterns that are not effectively exploited by other approaches. This finding is consistent with the neurobiological characteristics of disorders of consciousness: as shown by Achard et al. (2012), comatose patients exhibit a pronounced reorganization of hub structure and path-length–related properties, despite a largely preserved global network topology.

### 3.3. Computational and environmental costs

Classical machine-learning models and graph kernel methods were executed on the CPU of a Dell workstation running Ubuntu 22.04 and equipped with an Intel Core i7 processor, whereas graph neural network (GNN) models were trained using an NVIDIA RTX A500 GPU on the same machine. Because GPU acceleration constitutes a fundamentally different computational regime, raw wall-clock runtimes obtained from CPU-based and GPU-based implementations should not be interpreted as direct algorithmic comparisons. Moreover, GPU-accelerated training typically entails higher power consumption, potentially leading to increased environmental costs. Accordingly, we report in Table 3 the computational runtime and $CO_2$ emissions exclusively for classical machine-learning and graph kernel methods; $CO_2$ emissions were estimated using CodeCarbon (Courty et al., 2024b).

### 4. Conclusion

In this work, we conducted a comprehensive benchmarking study of classical machine-learning methods, graph kernel approaches, and graph neural networks (GNNs) on four resting-state fMRI brain graphs across multiple classification tasks. By evaluating predictive performance, statistical significance, and computational cost within a unified experimental framework, we provide a principled assessment of the practical trade-offs between model families. Across the evaluated datasets, well-tuned classical machine-learning methods and graph kernels achieve performance comparable to GNN architectures. In particular, for

small clinical datasets, classical approaches such as logistic regression, random forests, and kernel-based methods exhibit strong predictive accuracy while incurring substantially lower computational and environmental costs. Although GNNs are often presented as the dominant paradigm for graph-structured neuroimaging data, our results indicate that their empirical advantages over classical baselines are limited and strongly task-dependent. Several limitations should be acknowledged. First, our analysis relies on AAL-based parcellations with fixed node ordering across datasets, which ensures a fair comparison across model families by preserving consistent anatomical correspondence. However, this design choice may influence the relative performance of methods, particularly as classical approaches are not permutation-invariant. As a result, the extent to which our conclusions generalize to alternative parcellation schemes, especially those involving a larger number of regions, remains an open question. Second, for functional connectivity estimation, we focus in this work on Pearson correlation, as it remains the most widely adopted measure in the literature. Nevertheless, we fully acknowledge that Pearson correlation is not necessarily the optimal estimator of brain connectivity. In parallel work building upon distribution-based connectivity estimators (Lbath et al., 2024), we investigate richer representations of inter-regional variability, which have shown promising advantages for brain graph modeling (Mhanna et al., 2026). Third, while we relied on well-established hyperparameter configurations informed by prior large-scale benchmarks (Cui et al., 2022) and recent studies (Comparini et al., 2026), we acknowledge that dataset–specific tuning of GNNs could yield modest performance improvements. However, such gains are unlikely to fundamentally alter the relative positioning of GNNs with respect to classical baselines in the fixed-parcellation, static connectome setting considered here, and must be balanced against the associated substantial computational costs. Overall, our findings question the assumption that increased model complexity necessarily leads to superior performance in brain graph analysis. In line with recent calls for responsible and reproducible machine learning, we emphasize the importance of transparent reporting, fair baselines, and cost–aware evaluation in future applications of graph learning methods to neuroimaging.

## Acknowledgments

This work was supported by the Agence Nationale de la Recherche under the France 2030 programme, reference ANR-23-IACL-0006.

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
