# OpenReview forum: "A comprehensive benchmark of graph neural networks, graph kernels, and classical machine learning approaches on rs-fMRI brain graphs"
_MIDL.io/2026/Validation_Papers — MIDL 2026 - Validation Papers Poster_

### Official Review · Reviewer_wJqr · 2025-12-16

**Confidence:** 4
**Preliminary Rating:** 4
**Final Rating:** 4

**Summary:**

This paper presents a large-scale, systematic benchmark comparing classical machine-learning models, graph kernel methods, and graph neural networks for resting-state fMRI brain graph classification. Experiments are conducted across 4 heterogeneous datasets, using a rigorous repeated nested cross-validation protocol with corrected statistical testing. The results show that well-tuned classical models and graph kernels often match or outperform GNNs, particularly on small or heterogeneous datasets, while being substantially more computationally efficient.

**Strengths:**

1. This paper evaluates multiple model families under a unified and carefully designed experimental framework, thereby directly addressing the fragmentation issues present in previous benchmarking efforts.
2. The authors release the code, hyperparameter and seed setting for their experiments, which is exemplary for reproducibility.
3. The authors report the training time and CO₂ emissions for classical and kernel methods, which adds a valuable sustainability and deployment perspective that is often missing from benchmarking studies.

**Weaknesses:**

1. GNN models are evaluated using fixed hyperparameters adopted from prior studies. While computationally pragmatic, this may disadvantage GNNs relative to classical methods that receive extensive tuning, potentially affecting fairness of comparison.
2. The study relies on Pearson correlation–based functional connectivity with a single atlas choice (AAL variants). Other connectivity measures (e.g., partial correlation, tangent space) and parcellations could influence relative model rankings.
3. The benchmark focuses exclusively on predictive performance and cost. Interpretability, learned representations, or neurobiological plausibility of model outputs are not examined.

**Detailed Comments:**

1. Clarify whether GNN hyperparameters were validated on any subset to ensure they are reasonably competitive.
2. The authors could discuss how the results might change when using different connectivity estimation methods or graph resolutions. Perhaps including some experiments using alternative brain parcellation schemes would make the conclusions of this paper more convincing.
3. The authors might consider adding a brief section on limitations, outlining the limitations of graph construction, model scope, and dataset, which would help improve the clarity of the paper.

**Justification Of Final Rating:**

This paper delivers a rigorous, transparent, and reproducible benchmark with clear practical implications. The statistical rigor, breadth of evaluation, and emphasis on reproducibility and efficiency make this work a valuable contribution to the community. However, the authors still did not resolve my concerns regarding GNN tuning and graph construction choices. As a result, my final rating still stands.

**Justification Of The Preliminary Rating:**

This paper delivers a rigorous, transparent, and reproducible benchmark with clear practical implications. The statistical rigor, breadth of evaluation, and emphasis on reproducibility and efficiency make this work a valuable contribution to the community. Minor concerns regarding GNN tuning and graph construction choices do not outweigh the overall strength and utility of the study.

**Questions To Address In The Rebuttal:**

1. In this article, the author only retained the top 10% of the strongest connections to construct the adjacency matrix. Have the authors evaluated whether the relative ranking of classical ML, graph kernels, and GNNs remains stable under alternative connectivity measures (e.g., partial correlation, tangent-space embedding) or different sparsity levels?
2. Since during the experiment, all datasets use AAL-based parcellations with fixed node ordering. To what extent do the authors believe their conclusions depend on this fixed-node assumption?
3.According to Sec 2.5, classical models and graph kernels are extensively tuned via Bayesian optimization, while GNNs use fixed hyperparameters adopted from prior work.Is it possible that part of the observed performance gap reflects under-tuning of deep models rather than inherent model limitations?
4. The COMA dataset contains only 44 subjects, yet is evaluated with repeated nested cross-validation. How stable are fold-level predictions in this regime, and how often do the same subjects appear in decisive test splits?

---

### Official Review · Reviewer_5LFF · 2025-12-22

**Confidence:** 4
**Preliminary Rating:** 4

**Summary:**

This paper presents a large-scale, carefully designed benchmark comparing classical machine-learning models, graph kernels, and graph neural networks for classification tasks on resting-state fMRI brain graphs. The authors evaluate these methods across four heterogeneous cohorts (HCP, ABIDE, ADHD, and COMA; N=1513) using a rigorous repeated nested cross-validation protocol and corrected statistical significance testing. The study shows that, for relatively small brain graphs with fixed vertex ordering, well-tuned classical models and graph kernels often match or outperform GNNs while requiring substantially lower computational and environmental costs. By releasing all code, folds, seeds, and logs, the paper provides a transparent and reproducible reference point for model selection in connectome-based learning. Overall, the work is significant as a validation study that challenges the assumption that increased model complexity necessarily leads to superior performance in rs-fMRI graph analysis.

**Strengths:**

The main strength of the paper lies in its methodological rigor and transparency. The use of repeated nested cross-validation combined with corrected repeated k-fold statistical tests and FDR correction provides unusually reliable model comparisons for neuroimaging data. The inclusion of multiple heterogeneous datasets and tasks strengthens the generality of the conclusions, while the unified experimental pipeline enables fair comparisons across model families. The benchmark is well positioned with respect to prior work, clearly building on and extending both BrainGB and the classical benchmark of Dadi et al. (2019). The detailed reporting of computational time and CO emissions adds practical value and aligns with current concerns about sustainable machine learning. As a validation paper, it offers high utility to the community despite not proposing a novel model.

**Weaknesses:**

The study is limited to relatively small graphs with fixed node ordering and unimodal rs-fMRI connectivity, which constrains the scope of the conclusions and may not generalize to higher-resolution parcellations, dynamic connectivity, or multimodal graphs. GNN hyperparameters are largely fixed rather than tuned per dataset, which may disadvantage deep models relative to classical methods, even if this choice is justified by computational cost. The COMA dataset is very small, and although handled carefully, its results show high variance and should be interpreted cautiously. In addition, the benchmark focuses exclusively on classification performance and does not consider interpretability, uncertainty estimation, or downstream clinical utility, which are increasingly relevant in neuroimaging applications.

**Detailed Comments:**

The paper would benefit from a short discussion on how results might change with higher-resolution atlases or dynamic functional connectivity. Clarifying whether the fixed GNN hyperparameters were validated on any dataset would improve transparency. A brief qualitative error analysis, especially for COMA and ADHD, could help contextualize the observed performance variability. Explicitly stating whether GPU energy consumption was excluded by design would further clarify the environmental cost analysis.

**Justification Of The Preliminary Rating:**

This paper makes a solid and timely contribution as a validation and benchmarking study rather than a methodological advance. Its value lies in correcting common experimental weaknesses in prior neuroimaging benchmarks, particularly around cross-validation, statistical testing, and reproducibility. While the work is limited in scope to static rs-fMRI graphs and does not fully optimize deep models, the conclusions are well supported by the evidence presented. The benchmark provides clear, actionable guidance to researchers and practitioners, which justifies acceptance despite the acknowledged limitations.

**Questions To Address In The Rebuttal:**

How sensitive do the authors expect the relative ranking of GNNs versus classical methods to be if dataset-specific hyperparameter tuning were allowed for GNNs?
Do the authors anticipate similar conclusions for larger graphs or dynamic functional connectivity representations, or are the results primarily tied to fixed-order, static connectomes?

---

### Official Review · Reviewer_31jR · 2025-12-27

**Confidence:** 4
**Preliminary Rating:** 3
**Final Rating:** 4

**Summary:**

This study introduces a comprehensive benchmarking framework across four heterogeneous cohorts  and multiple classification tasks. The authors systematically evaluate classical machine-learning models, graph kernel methods, and several representative graph neural network architectures. Overall, this type of large-scale, statistically grounded benchmarking study is highly valuable and timely for the field.

**Strengths:**

The study leverages a sufficiently large multi-cohort dataset and focuses on a topic of broad interest within brain network analysis. The benchmarking is comprehensive, covering a wide spectrum of modeling approaches, and the evaluation protocol is carefully designed, with appropriate statistical testing to support the conclusions. The inclusion of quantitative significance analysis strengthens the reliability of the reported findings.

**Weaknesses:**

While the overall motivation is sound, I have several concerns that limit the impact of the work.

1. The study centers heavily on GNNs; however, I am not fully convinced that GNNs constitute an ideal backbone for brain network modeling. GNNs primarily emphasize node representation learning under edge-constrained message passing, whereas the core object of interest in brain network analysis is the connectivity itself. Despite the large body of recent GNN-based literature, from my perspective many such applications remain conceptually misaligned with the neuroscience problem. In fact, the results in this paper appear to support this concern, as several GNN methods perform no better, and sometimes worse, than classical machine-learning approaches. This does not diminish the technical quality of the study, but it does somewhat reduce its impact, as the choice of GNNs may not be the most informative modeling direction. Extending the framework to include transformer-based or attention-driven architectures in future work is suggested.

2. Some of the compared GNN architectures appear to be relatively outdated, and the study would benefit from including more recent or expressive models to better reflect the current state of the field.

3. Given that the core innovation of GNNs lies in message passing and aggregation mechanisms, I strongly suggest expanding the discussion and analysis around how these components influence performance. A more detailed examination of aggregation strategies or message-passing depth would improve interpretability and help contextualize the observed results.

4. The paper primarily discusses GNNs in a general sense, but the evaluated models are essentially inductive GNNs (graph classification). Transductive approaches (node classification), such as population-level GNNs, are not well suited to the problem setting considered here. I recommend explicitly clarifying this distinction in the manuscript to avoid potential confusion or misinterpretation by readers.

**Detailed Comments:**

See above.

**Justification Of Final Rating:**

The authors have addressed several of my concerns. Compared to the initial version, the subsequent updates have substantially increased the value of the paper. Based on these improvements, I am willing to raise my score.

**Justification Of The Preliminary Rating:**

This is an interesting and well-motivated study, but the overall impact is somewhat limited by the use of outdated comparison methods, and by the diminishing influence of GNN-based approaches in this research area.

**Questions To Address In The Rebuttal:**

I would be willing to raise my score if the authors adequately address points (3) and (4) in weakness

---

### Author Rebuttal · Authors · 2026-01-24

**Rebuttal:**

We have carefully addressed all reviewers’ comments and suggestions in the revised version of the manuscript. All modifications are highlighted in red in the revised PDF for clarity.

**Supporting Material:**

/attachment/cd2eca020e289959ec842d7c0f9c8d0406f4af1c.pdf

---

### Meta-Review · Area_Chair_s4UZ · 2026-02-05

**Recommendation:** Accept (Poster)
**Confidence:** 4

**Metareview:**

This is a well-done benchmark with a cross‑validated evaluation and statistical testing across four rs‑fMRI cohorts. Given the good methodological quality, transparent code/data release, and practical relevance for connectome‑based modeling, I recommend acceptance.

---

### Decision · Program_Chairs · 2026-02-14

Accept (Poster)